# Fusogenic structural changes in arenavirus glycoproteins are associated with viroporin activity

You Zhang[1,2], Joanne York[3], Melinda A. Brindley[4], Jack H. Nunberg[3], Gregory B. Melikyan[1,2]*

1 Department of Pediatrics, Emory University School of Medicine, Atlanta, Georgia, United States of America, 2 Children's Healthcare of Atlanta, Atlanta, Georgia, United States of America, 3 Montana Biotechnology Center, University of Montana, Missoula, Montana, United States of America, 4 Department of Infectious Diseases, Department of Population Health, College of Veterinary Medicine, University of Georgia, Athens, Georgia, United States of America

* gmeliki@emory.edu

**Data Availability Statement:** All relevant data are within the manuscript and its Supporting Information files.

## Abstract

Many enveloped viruses enter host cells by fusing with acidic endosomes. The fusion activity of multiple viral envelope glycoproteins does not generally affect viral membrane permeability. However, fusion induced by the Lassa virus (LASV) glycoprotein complex (GPc) is always preceded by an increase in viral membrane permeability and the ensuing acidification of the virion interior. Here, systematic investigation of this LASV fusion phenotype using single pseudovirus tracking in live cells reveals that the change in membrane barrier function is associated with the fusogenic conformational reorganization of GPc. We show that a small-molecule fusion inhibitor or mutations that impair viral fusion by interfering with GPc refolding into the post-fusion structure prevent the increase in membrane permeability. We find that the increase in virion membrane permeability occurs early during endosomal maturation and is facilitated by virus-cell contact. This increase is observed using diverse arenavirus glycoproteins, whether presented on lentivirus-based pseudoviruses or arenavirus-like particles, and in multiple different cell types. Collectively, these results suggest that conformational changes in GPc triggered by low pH and cell factor binding are responsible for virion membrane permeabilization and acidification of the virion core prior to fusion. We propose that this viroporin-like activity may augment viral fusion and/or post-fusion steps of infection, including ribonucleoprotein release into the cytoplasm.

## Author summary

The fusion of enveloped virus with host cell membranes is mediated by extensive conformational changes in viral glycoproteins, triggered by binding to cognate receptors and/or by exposure to acidic pH in the maturing endosome. We have previously reported that, unlike many other viral glycoproteins, pH-triggered endosomal fusion by the Lassa virus viral glycoprotein complex (GPc) is preceded by a mild permeabilization of the viral membrane. Here, we provide evidence that this activity is associated with early fusogenic

**Funding:** This work is supported by Division of Intramural Research, National Institute of Allergy and Infectious Diseases; NIH R01 AI053668 to GBM and NIH R01 AI139238 to MAB. The funders had no role in study design, data collection and analysis, decision to publish, or preparation of the manuscript.

changes in the GPc and is markedly enhanced by virus-cell contact. Permeabilization is induced by the glycoproteins of diverse arenaviruses and occurs in multiple target cell types. Based on these observations, we propose that, by analogy to the influenza virus M2 channel, membrane permeabilization and the resultant acidification of the arenavirus interior facilitate viral fusion and/or post-fusion steps of infection.

## Introduction

Arenaviruses initiate infection by entering cells *via* endocytosis and fusion of their envelope membrane with the endosomal membrane. Arenavirus-cell fusion is mediated by the trimeric viral glycoprotein complex (GPc), each protomer of which consists of three noncovalently associated subunits, GP1, GP2, and a stable signal peptide (SSP) [1–5]. The membrane-distal GP1 subunit engages host attachment factors and receptors [4,6–14], while conformational changes in the transmembrane GP2 subunit [15] drive the merger of viral and cellular membranes. These fusogenic changes in GPc are triggered by low endosomal pH and may be augmented by interactions with endosome-resident viral co-receptors. For example, the Old-World Lassa arenavirus (LASV) engages α-dystroglycan on the cell surface and, upon entry into acidic endolysosomes, switches to the LAMP1 receptor [16–19].

We have developed labeling and imaging strategies for robust single virion tracking and detection of individual HIV-1 pseudovirus fusion events. These strategies include incorporation of an HIV-1 protease-cleavable Vpr construct that produces free mCherry, a viral content marker, and a pH-sensitive YFP-Vpr protein packaged into the mature viral core, to detect changes in pH [20]. With this labeling approach, virus-cell fusion is manifested by a rapid release of mCherry through the fusion pore, whereas the viral core-associated YFP signal persists, aiding reliable detection of single virus fusion events [20]. Using this HIV-1 pseudovirus platform, we have studied entry driven by a variety of viral glycoproteins, including the Influenza A virus (IAV) HA, Vesicular Stomatitis Virus (VSV) G and Avian Sarcoma and Leukosis Virus (ASLV) Env [21–31], all of which mediate virus entry from acidic endosomes.

Recent investigations of the LASV entry mechanisms using HIV-1 pseudoparticles (LASVpp) revealed an unusual fusion phenotype. In contrast to "tight" fusion induced by other viral glycoproteins, LASVpp fusion is preceded by viral membrane permeabilization [31,32]. The increase in membrane permeability to protons is detected as quenching of pH-sensitive YFP-Vpr fluorescence. Subsequent virus-endosome fusion results in a loss of the free mCherry signal due to its release into the cytoplasm, and simultaneous recovery of the YFP signal reflecting re-neutralization of the viral interior [31,32]. Analysis of the temporal relationship between changes in the intraviral pH and mCherry release allowed us to delineate the initial dilation of fusion pores. We have demonstrated that binding to human LAMP1, while not strictly required for LASV fusion, strongly promotes the enlargement of the nascent fusion pore [32]. However, the mechanism of viral membrane permeabilization and the potential role of the virion interior acidification in LASV entry/infection remain unclear.

Here, using an HIV-1 based pseudovirus platform and arenavirus-like particles (VLPs), we demonstrate that viral membrane permeabilization is a common feature of fusion mediated by GPc of diverse arenaviruses and in different cell types. We also show that the increase in viral membrane permeability is mediated by GPc upon endosome acidification and is augmented by virus contact with the cell. Importantly, genetic, and pharmaceutical interventions that impair LASV GPc's fusion activity diminish its ability to increase membrane permeability, suggesting that functional refolding of the arenavirus glycoprotein compromises the barrier

function of the viral membrane. By analogy to the Influenza virus M2 channel function [33,34], we suggest that the increase in viral membrane permeability and the ensuing acidification of the arenavirus interior may be important for post-fusion events in LASV entry.

## Results

### Viral membrane permeabilization is an intrinsic feature of arenavirus GPc-mediated fusion

We have previously reported that fusion of Lassa GPc-pseudotyped HIV-1 particles (LASVpp) with the endosome is associated with an increase in virion membrane permeability and acidification of the virion interior, regardless of whether membrane fusion is triggered by endosomal acidification or exposure to low pH at the cell surface [31,32]. Permeabilization of the virion membrane consistently occurs prior to formation of the fusion pore. In these studies, membrane permeabilization and viral fusion events were detected by labeling the pseudovirus with mCherry-CL-YFP-Vpr, which is cleaved by the HIV-1 protease at the cleavage site (CL) to generate free mCherry, and with YFP-Vpr, which remains associated with the virion core (Fig 1A) [20]. Permeabilization of the viral membrane and acidification of the virion interior are manifested by quenching of the pH-sensitive YFP signal; free mCherry is too large to diffuse out of the permeabilized virions. Subsequent formation of the fusion pore leads to a quick recovery of YFP fluorescence due to re-neutralization of viral interior and loss of mCherry signal due to release into the cytoplasm (Fig 1A and 1B and S1 Movie) [20,32]. This phenomenon is in contrast with the "tight" fusion of viral and endosomal membranes promoted by other viral glycoproteins, which is not associated with YFP quenching (S1 Fig and S2 Movie). Since it is unclear whether acidification of the virion interior occurs through defects forming in the viral lipid membrane or through a channel-like proteinaceous pore, we will operationally refer to this phenomenon as pre-fusion viral membrane permeabilization (PVMP).

To determine the generality of GPc-mediated viral membrane permeabilization, fusion of single HIV-1 particles pseudotyped with the glycoproteins of another Old World arenavirus (LCMVpp) or distantly related New World arenaviruses (MACVpp and JUNVpp) was analyzed. All these pseudoviruses reproducibly underwent PVMP in acidic endosomes (Figs 1C–1E and S2). We next examined LASVpp entry into different cell types. Our results (Fig 1D and 1E) show that PVMP during LASVpp fusion is cell type-independent, as it consistently occurs in unrelated human and simian cells lines. The efficiency of such LASVpp fusion (percent of cell-bound particles that release mCherry after YFP quenching) varied somewhat depending on the cell line, with U2OS cells being the most permissive and VeroE6 cells being the least permissive (Fig 1D). Interestingly, the rates of LASVpp and MACVpp fusion were not significantly different from each other but were considerably slower than that of IAVpp (Fig 1E).

To exclude the possibility that PVMP might be an artefact of HIV-1 pseudotyping with arenavirus GPc, we generated arenavirus-like particles (VLPs) comprising the Candid-1 (JUNV vaccine strain [35,36]) matrix (Z) and nucleoprotein (NP) proteins and bearing GPcs of diverse arenaviruses, essentially as described in [37]. In order to track VLPs and detect viral membrane permeabilization and fusion, we fluorescently labeled NP by replacing the D93 residue in an exposed loop [38] with the YFP sequence, for sensing intraviral pH, or with the pH-insensitive mCherry sequence (referred to as NP-DYFP and NP-DmCherry, respectively, Fig 2A). Virions produced using a mixture of NP-DYFP and NP-DmCherry are positive for both markers. Unlike the HIV-1 based pseudoviruses labeled with cleavable mCherry-CL-YFP-Vpr (Fig 1), the tagged NP proteins are not subject to cleavage and remain associated with the viral ribonucleoprotein complex after fusion (Fig 2A). Using this VLP system, acidification of the virion interior and formation of the fusion pore are detected through YFP quenching and

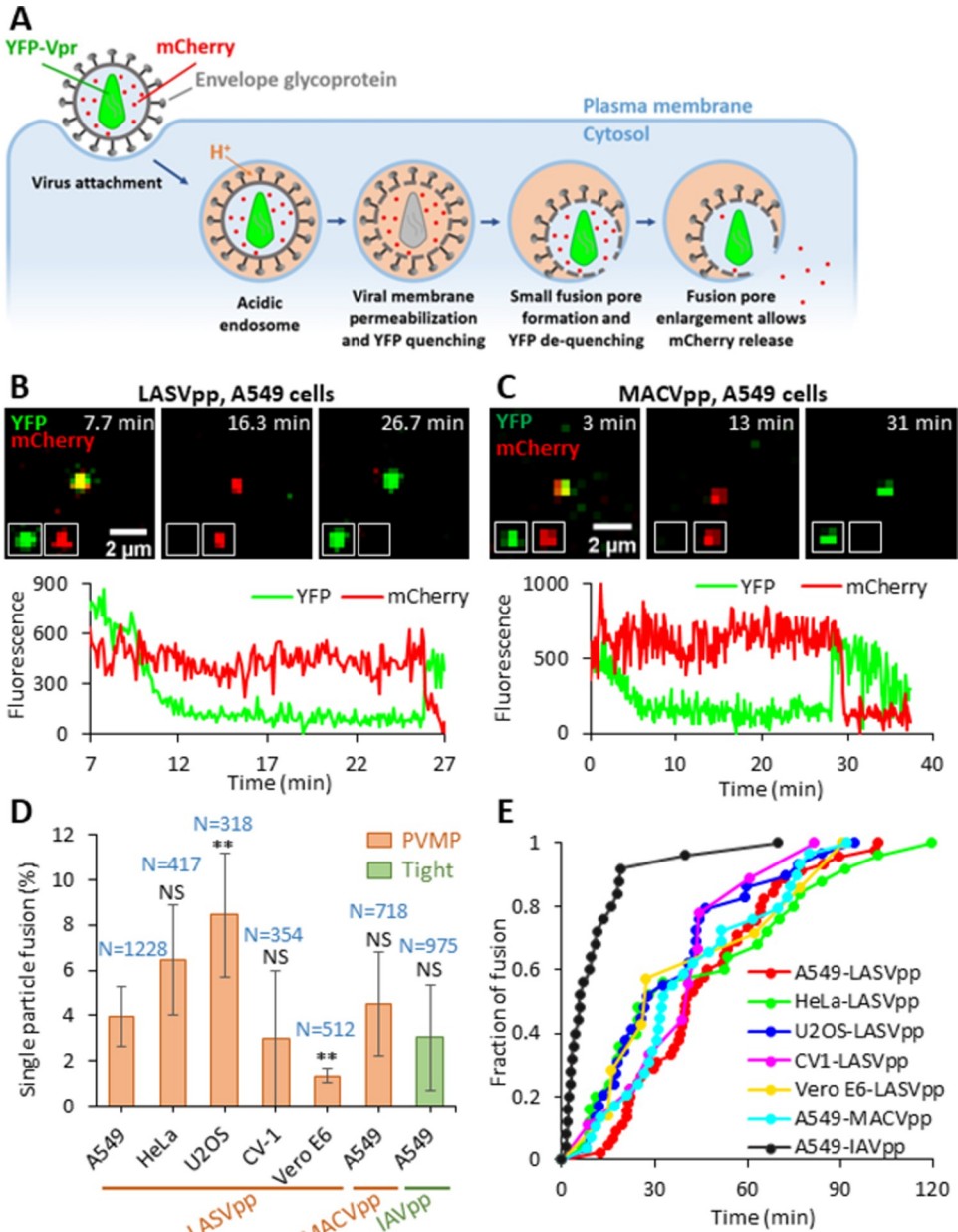

**Fig 1. Arenavirus membrane permeability increases prior to virus fusion.** (A) Illustration of mCherry-CL-YFP-Vpr labeled single LASVpp fusion. LASVpp is internalized and trafficked to acidic endosomes where the viral membrane is permeabilized. Increases in viral membrane permeability lead to acidification of the virus' interior which is manifested in YFP signal quenching. LASVpp-endosome fusion results in mCherry release into the cytoplasm and concomitant re-neutralization of the virus' interior, seen as recovery of YFP signal. (B) Single LASVpp fusion with A549 cell. Time-lapse images (top) and fluorescence traces (bottom) show virus interior acidification (YFP quenching) at 12.3 min and fusion (YFP dequenching and mCherry loss) at 25.9 min (see S1 Movie). (C) A single MACVpp fusion event in A549 cell. Time-lapse images (top) and fluorescence traces (bottom) show virus interior acidification (YFP quenching) at 7.0 min and fusion (YFP dequenching and mCherry loss) at 29.0 min. (D) HIV-1 particles labeled with mCherry-CL-YFP-Vpr and pseudotyped with LASV, MACV GPc or IAV HA were attached to cells by spinoculation in the cold, and their entry/fusion was triggered by shifting to 37˚C. Percentage of particles releasing mCherry is plotted. Data are means ± SD of 3 independent experiments. Results were analyzed by Student's t-test. Numbers on the top of bars are numbers of total particles analyzed. Asterisks and NS on the top of bars represent the significance relative to the LASVpp fusion efficiency in A549 cells. **, p<0.01; NS, not significant. (E) Kinetics of single GPc and IAV pseudovirus fusion with different target cells.

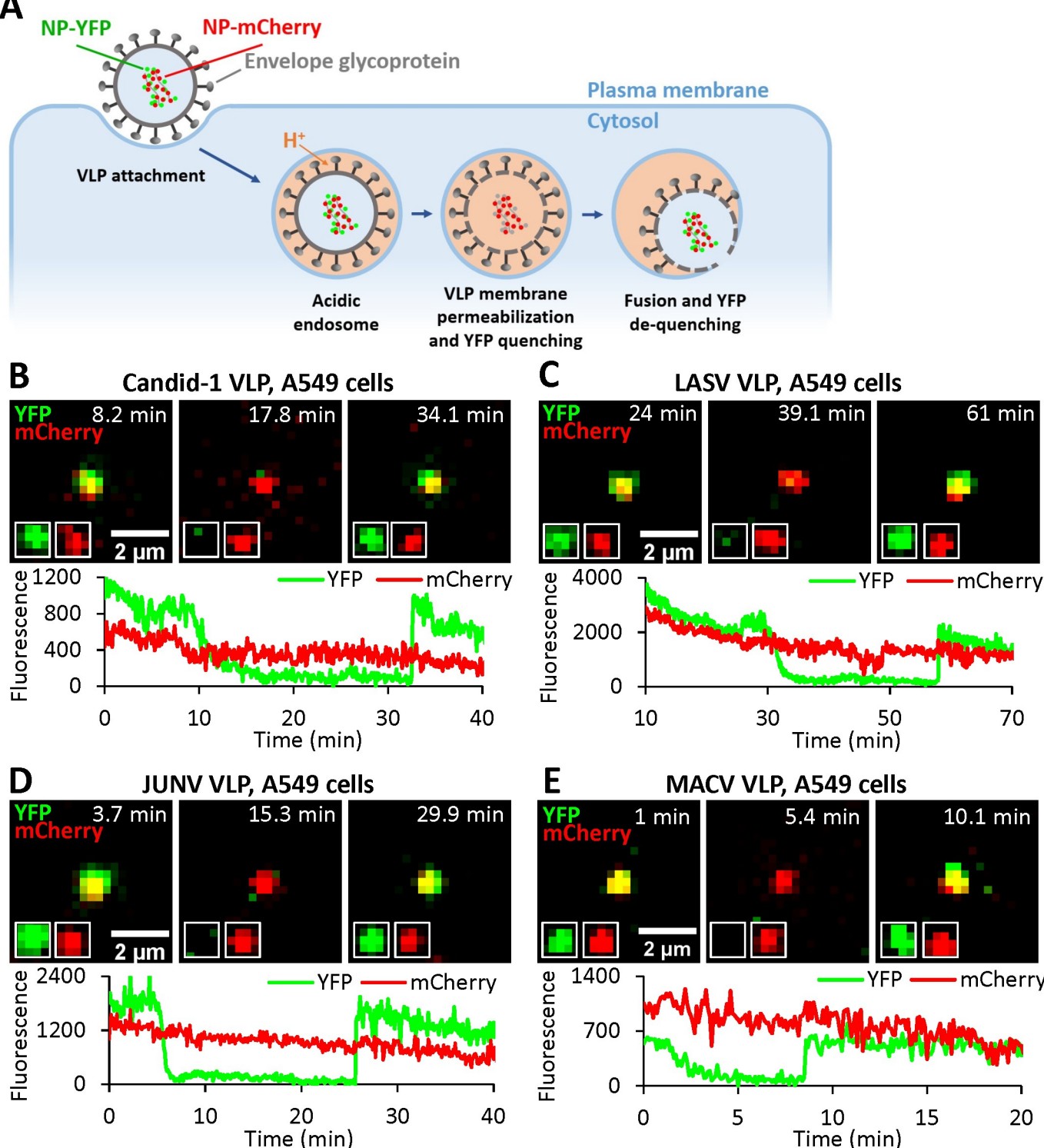

**Fig 2. Arenavirus VLPs undergo membrane permeabilization prior to fusion.** (A) Illustration of single VLP fusion labeled with NP-DmCherry/NP-DYFP. VLP is internalized, trafficked to acidic endosomes where the viral membrane is permeabilized, leading to acidification of the viral interior and quenching of YFP fluorescence. Subsequent fusion of VLP and endosomal membrane neutralizes the VLP interior and results in recovery of YFP signal. (B) A single Candid-1 VLP fusion event in A549 cell. Time-lapse images (top) and fluorescence traces (bottom) show VLP interior acidification (YFP quenching) at 12.1 min and fusion (YFP dequenching) at 33.4 min. (see S3 Movie). (C) A single LASV VLP fusion event in A549 cell. Time-lapse images (top) and fluorescence traces (bottom) show virus interior acidification (YFP quenching) at 34.0 min and fusion (YFP dequenching) at 57.9 min. (D) A single JUNV VLP fusion event in A549 cell. Time-lapse images (top) and fluorescence traces (bottom) show VLP interior acidification (YFP quenching) at 7.3 min and fusion (YFP dequenching) at 25.6 min. (E) A single MACV VLP fusion event in A549 cell. Time-lapse images (top) and fluorescence traces (bottom) show virus interior acidification (YFP quenching) at 4.0 min and fusion (YFP dequenching) at 8.7 min.

subsequent recovery of the YFP signal upon re-neutralization, respectively (Fig 2B–2E). The pH-insensitive mCherry signal serves as reference for ensuring that YFP fluorescence changes are not due to particle departure from a focal plane.

This VLP labeling approach allowed us to detect single particle fusion events mediated by Candid-1, LASV, JUNV and MACV GPcs (Fig 2B–2E). All arenavirus glycoproteins tested reproducibly increase VLP membrane permeability, seen as NP-DYFP quenching, and mediate virus fusion with endosomes, as evidenced by NP-DYFP signal dequenching. Thus, viral membrane permeabilization prior to arenavirus GPc-mediated fusion pore formation is independent of the pseudotype platform, occurring in both HIV- and arenavirus-based particles. We noticed differences in the extents and rates of fusion of VLPs pseudotyped with distinct arenavirus glycoproteins. In the absence of a pan-arenavirus GPc monoclonal antibody, we were unable to directly compare levels of GPc incorporation into the various VLPs and thus could not exclude that such differences may contribute to the observed differences in the extent and kinetics of fusion among the arenavirus VLPs. In contrast, no virus membrane permeability increase was observed upon entry of VLPs bearing the VSV-G protein (S3A Fig), despite its incorporation into VLPs (S3C-S3E Fig).

Together, the above results support the notion that arenavirus membrane permeabilization and the resulting acidification of the virion interior may be an obligatory intermediate step of arenavirus GPc-mediated fusion.

## Permeabilization of arenavirus membrane occurs shortly after exposure to low pH

GPc-mediated membrane fusion can be triggered by exposure to acidic pH, independent of a cell surface receptor [7,39]. To investigate whether cell contact is necessary for virion membrane permeabilization, we exposed coverslip-adhered LASVpp to a pH 5.0 membrane-impermeable citrate buffer. Only 3.9% of particles exhibited immediate YFP quenching upon acidification, demonstrating that nearly all pseudoviruses have intact membranes that limit the diffusion of protons (S4 Fig). To assess the temporal relationship between endosome acidification and LASVpp membrane permeabilization, pseudoviruses were co-labeled with two pH-sensitive fluorescent proteins–HIV-1 Gag-ecliptic pHluorin (Gag-EcpH, intraviral sensor) and pDisplay-pHuji, an external pH-sensor that is anchored to the virion membrane by the transmembrane domain of platelet-derived growth factor receptor (Fig 3A) [40]. Pseudovirus entry into cells was synchronized by pre-binding dual-labeled particles to cells in the cold and quickly shifting to 37˚C. Upon endocytic uptake of the virus by the cell, endosomal acidification is observed as the loss in the pHuji signal, whereas LASVpp membrane permeabilization is seen as loss of the EcpH signal (Fig 3B). In contrast, HIV-1 particles pseudotyped with the Influenza virus HA (IAVpp) exhibited quenching of the surface marker (pHuji) without concomitant quenching of the internal sensor (EcpH, Fig 3C). Interestingly, despite the presence of GPc on virtually all pseudovirion particles (S5 Fig), only 14% of cell-bound particles exhibited a loss of pHuji signal within 2 hrs of infection (Fig 3D), suggesting that LASVpp entry into acidic compartments in A549 cells is inefficient and/or slow. As expected, loss of pHuji signal was potently inhibited by Bafilomycin A1 pretreatment, which blocks endosome acidification (Fig 3D). Importantly, the kinetics of single particle pDisplay-pHuji and Gag-EcpH quenching were similar (half-times 55.2 and 59.8 min, respectively, Fig 3E), implying that acidification of virus interior after exposure to low pH is faster than the time required for virus uptake and acidification of virus-carrying endosomes. Indeed, the average lag time between pHuji quenching (acidification of endosomal lumen) and subsequent loss of EcpH signal (acidification of the virion interior) was 0.8 min (Fig 3F).

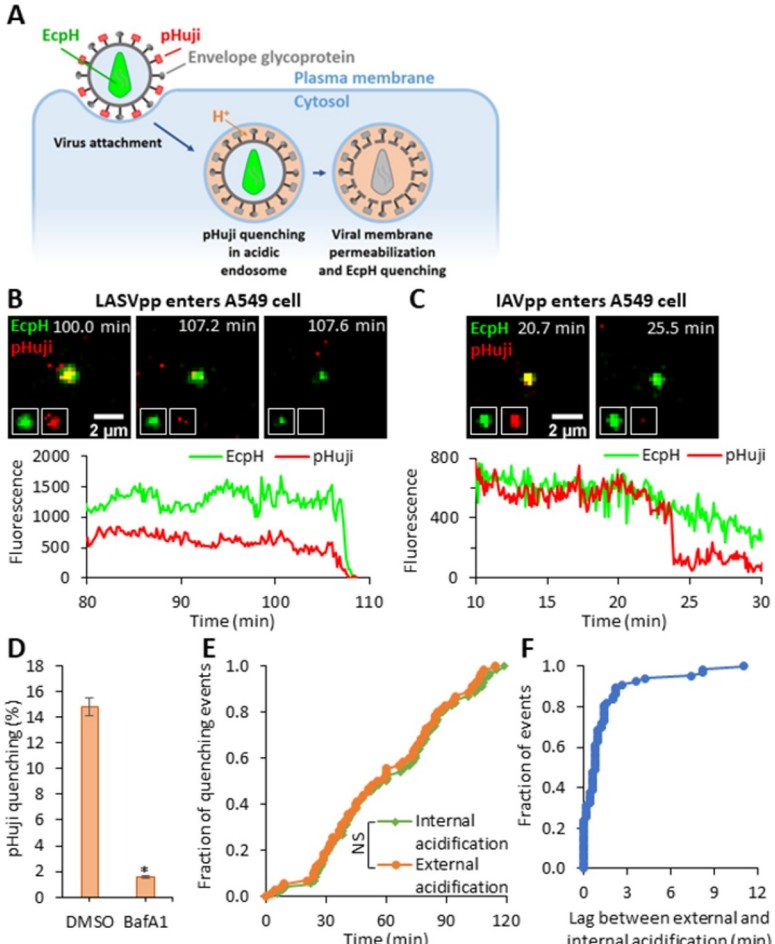

**Fig 3. LASV pseudovirus interior acidification occurs shortly after entering acidic compartments.** (A) Illustration of internalization and trafficking of LASVpp colabeled with pHuji and Gag-EcpH. LASVpp is internalized and trafficked to acidic endosomes where the viral surface probe, pHuji, is quenched. The LASVpp membrane permeability increases in acidic compartments, leading to the viral interior acidification and quenching of the internal low pH probe, EcpH. (B) Single LASVpp entry into acidic endosome and viral membrane permeabilization in A549 cells. Time-lapse images (top) and fluorescence traces (bottom) show that, shortly after virus entry into the acidic endosomes (pHuji quenching at 107.6 min), membrane permeabilization occurs, resulting in virus' interior acidification (EcpH quenching) at 108.2 min (see S4 Movie). (C) Single IAVpp entry into acidic endosome in A549 cells without membrane permeabilization. Time-lapse images (top) and fluorescence traces (bottom) show that IAVpp entry into the acidic endosome at 23.9 min leading to pHuji quenching with EcpH signal maintaining (see S5 Movie). (D) Bafilomycin A1 (BafA1) inhibits pHuji quenching. Data shown are means ± SD of 2 independent experiments. Results were analyzed by Student's t-test. *, p<0.05. (E) Kinetics of single LASVpp exterior and interior acidification (pHuji and EcpH quenching, respectively). Results were analyzed by Student's t-test. NS, not significant. (F) Distribution of lag times between pHuji and EcpH quenching for each single pseudovirus.

Together, our results show that the LASVpp membrane is permeabilized shortly after entering acidified endosomes. Such a short lag time to membrane permeabilization and high pKa values of pHuji ($\sim$7.7) and EcpH ($\sim$7.1) [40,41] suggest that virus membrane permeabilization occurs in early endosomes, soon after these compartments become mildly acidic.

## Cellular contact facilitates viral membrane permeabilization at low pH

To determine whether GPc interaction with cellular factors is required for viral membrane permeabilization at low pH, LASVpp labeled with mCherry-CL-YFP-Vpr were attached to

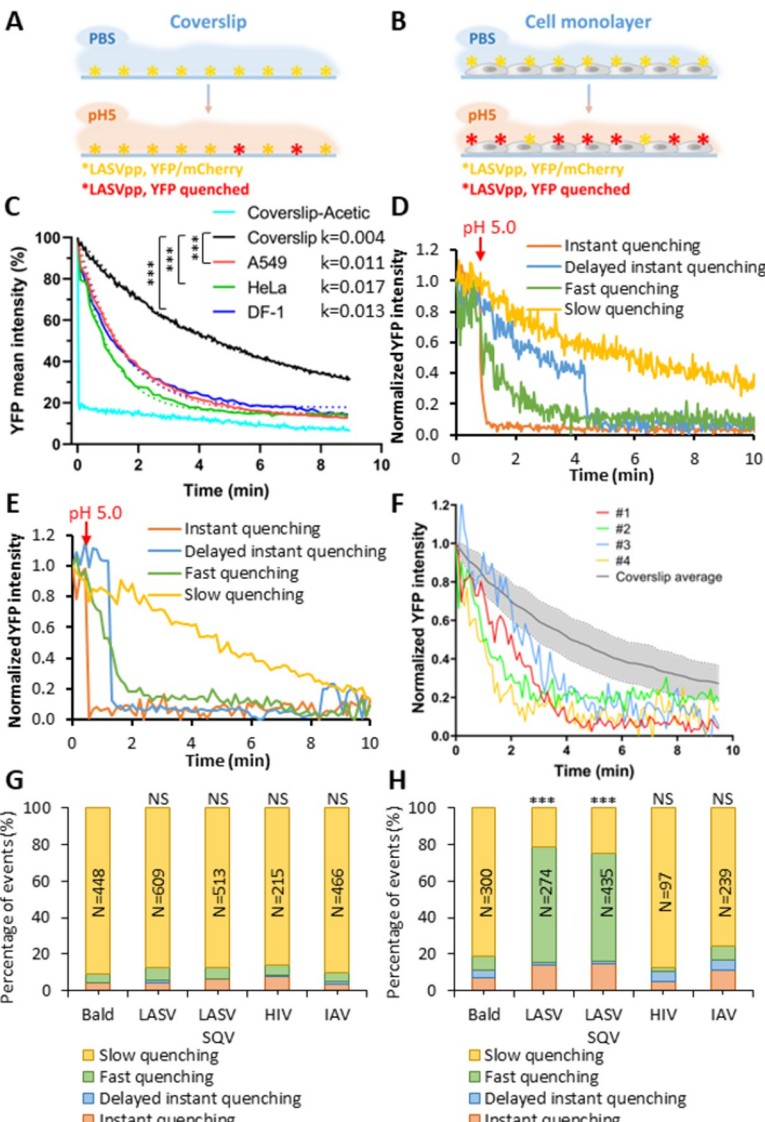

**Fig 4. Virus-cell contact promotes LASVpp membrane permeabilization at low pH.** (A, B) Illustration of the effects of LASVpp exposure to low pH on a coverslip (A) and on the cell surface (B). LASVpp were bound to the cell surface or to poly-L-lysine coated coverslips at 4˚C. GPc conformational changes are triggered by applying membrane impermeable pH 5.0 citrate buffer. (C) Mean YFP intensity decay of all coverslip-attached particles in the image field or on cells. Dotted-lines are single- exponential decay fits of data shown by solid lines. Control YFP quenching profile upon application of a membrane-permeable pH 5.0 acetic buffer is shown. Exponential decay rates k are in 1/sec. Results were analyzed by Student's t test, ***, p<0.001; NS, p>0.05. (D, E) Representative examples of four types of YFP quenching of single LASVpp on coverslip (D) or on cells (E). The point of adding a low pH buffer is marked with the red arrows. (F) Examples of the fast single virus YFP quenching events in A549 cell surface (#1-#4) overlaid onto the average YFP intensity profile for single LASVpp acidified on coverslip. The shaded area represents standard deviation of the mean YFP decay curve. (G, H) Quantification of the above four categories of YFP-Vpr quenching of single LASVpp on coverslip (G) or on DF-1 cells (H). Numbers within the bars are the total numbers of particles analyzed by Fisher's exact test. ***, p<0.001; NS, not significant.

coverslips, or to target cells in the cold to prevent virus uptake and fusion. Cell surface- or coverslip-bound viruses were then exposed to a membrane-impermeable pH 5.0 citrate buffer at 37˚C (Fig 4A and 4B). Analysis of loss of the total YFP-Vpr fluorescence for all mCherry-labeled particles in the image field revealed an exponentially decaying signal that was markedly

accelerated for cell-bound viruses compared to those attached to coverslips (Figs 4C and S6). The rate of the virion acidification was independent of the target cell type, displaying similar YFP decay constants in A549, HeLa and DF-1 cells (0.011, 0.017 and 0.013 sec$^{-1}$, respectively, Fig 4C). By contrast, a nearly instant quenching of YFP signal from 100% of particles was observed when a membrane-permeable acetic acid-based buffer was applied to viruses (Fig 4C), supporting the existence of significant barrier for proton permeation through the viral membrane under basal conditions.

To understand the basis for the loss of total YFP signal from multiple viral particles at low pH (Fig 4C), we tracked the YFP-Vpr signal of single LASVpp on coverslips or on the cell surface upon shifting to pH 5.0. Single virus YFP quenching profiles were classified into 4 categories (Fig 4D–4F): (1) "instant" quenching of YFP signal to background level upon acidification which was observed in 3.9% of particles (S4 Fig) presumably representing preexisting membrane defects; (2) "delayed instant" quenching after an initial slow decay of fluorescence, also in a very small fraction of particles exhibiting a variable lag to marked membrane permeabilization; (3) "slow" quenching, likely resulting from the baseline proton permeability of the viral membrane; and (4) "fast" gradual quenching reflecting the increased viral membrane permeability. (Additional examples of these distinct YFP-Vpr quenching events on coverslips and cells are shown in S7 Fig). Single YFP quenching events were considered fast, if the fractional YFP intensity of a particle fell below a standard deviation from the average rate of slow YFP quenching for viruses on the coverslip (Fig 4F). It should be noted that, judging by the gradual loss of YFP signal over the course of minutes, fast YFP quenching events appear to report a rather moderate membrane permeability increase. From the kinetics of fast YFP quenching in single virions, we estimated the membrane permeability to be on the order of $4 \cdot 10^{-8}$ cm/sec, which is only an order of magnitude higher that the estimated permeability corresponding to slow YFP quenching events (S8 Fig).

Approximately 88% of coverslip-adhered LASVpp exhibited slow/baseline YFP-Vpr quenching (Fig 4D and 4G), likely reflecting background proton permeability of the viral membrane; only ∼7.2% exhibited fast YFP quenching after low pH application. In sharp contrast, cell-bound viruses exhibited a dramatic increase in the fraction of fast YFP quenching events (∼60%) and only ∼20% exhibited slow/baseline quenching (Fig 4H). Other YFP quenching categories (instant and delayed instant) did not exhibit significant changes between cell-free and cell-attached particles. Thus, LASVpp-cell contact markedly promotes PVMP increases compared to cell-free virions.

Our results imply that efficient virion membrane permeabilization requires conditions that are permissive for GPc-mediated membrane fusion: low pH and virus contact with cells. This notion is supported by findings that "bald" particles lacking any viral glycoprotein or pseudoviruses expressing HIV-1 Env or IAV hemagglutinin (HA) do not exhibit fast YFP quenching on coverslips or when attached to cells (Fig 4G and 4H). The invariant fraction of fast YFP quenching events for coverslip-attached and cell-bound HIV-1 Env and IAV HA pseudotyped particles was not due to poor incorporation of these glycoproteins into HIV-1 particles (S9 Fig). In contrast, all control viruses exhibit predominant slow quenching, supporting the notion that the slow YFP quenching rate for these viruses likely represents passive (baseline) proton permeability of the viral membrane. We therefore hypothesized that fast YFP quenching is likely mediated by LASV GPc. We also tested if maturation of the HIV-1 core, which is known to regulate the Env function [42–44], affects viral membrane permeability. Immature LASVpp produced in the presence of the HIV-1 protease inhibitor, Saquinavir (SQV), exhibited acid-induced permeability increases on cells and not on coverslips, similar to the changes observed for mature particles (Fig 4G and 4H). Thus, HIV-1 maturation does not affect the ability of GPc to compromise the membrane's barrier function.

Together, the above results show that virus-cell contact at low pH is needed for efficient viral membrane permeabilization by LASV GPc. To determine whether specific LASV GPc interactions with the cellular receptors, α-dystroglycan or LAMP1, can promote the increases in viral membrane permeability, coverslip-attached pseudoviruses were pretreated with recombinant α-dystroglycan or a soluble fragment of LAMP1 (sLAMP1) and exposed to a low pH buffer supplemented with these proteins. Under these conditions, neither α-dystroglycan nor sLAMP1 accelerated the loss of YFP fluorescence at low pH (S10 Fig). This negative result may reflect the fact that these soluble receptors do not replicate effects of binding to the authentic membrane-bound proteins. Furthermore, cholesterol in the endosomal membrane appears to be required for functional LASV GPc interaction with LAMP1 [45].

## The increase in LASVpp membrane permeability is caused by fusion-inducing conformational changes in GPc

Having established that virion membrane permeabilization is mediated by LASV GPc upon exposure to low pH and contact with the target cell membrane, we next examined the link between the membrane permeability increases and virus fusion. Our results show that only ∼14% of all A549 cell-bound LASVpp are internalized into acidic compartments (Fig 3D) and that ∼30% of these particles fuse within 2 hrs of infection (Fig 1D and [31,32]). We tracked randomly selected single-pseudovirus YFP-Vpr quenching events for 2 hrs and categorized these depending on whether particles did or did not undergo membrane fusion (as judged by release of mCherry). We then plotted the ensemble average profiles of single particle YFP-Vpr quenching events (aligned to the onset of quenching) and determined the decay constants for fusing and non-fusing LASVpp (Fig 5A and 5B). Both the average profiles and individual rate constants of fusing particles showed a highly significant enhancement of membrane permeabilization relative to non-fusing particles. Approximately 80% of fusing LASVpp exhibited fast YFP-Vpr quenching, in contrast to only ∼20% in non-fusing particles (Fig 5C). Consistent with fusion-related increases in LASVpp membrane permeability, non-fusing particles exhibited similar YFP quenching profiles to particles attached to coverslips, with no detectable changes in the baseline proton permeability after exposure to low pH (Fig 5A and 5B).

To further investigate the relationship between the fusogenic conformational changes in GPc and virion membrane permeabilization, we examined a series of GPc mutants that are defective in fusion, as well as the effects of a small-molecule fusion inhibitor. For these studies, we utilized a virus-cell fusion assay, which is based on the cytosolic delivery of the pseudo-virus-incorporated β-lactamase-Vpr construct [46,47]. The K33A mutation lies within SSP, in a GPc region that appears to be involved in pH sensing and the activation of membrane fusion [48,49]. The fusion-defective K33A GPc [49] was efficiently incorporated into pseudovirions (see S11 Fig). LASVpp bearing this mutant GPc produced a near background level of fusion signal upon exposure of cell-attached pseudovirions to low pH (Fig 6A). Parallel imaging experiments also revealed a significant defect in the ability of K33A GPc to increase membrane permeability (Fig 6B). Accordingly, the fraction of the fast single-virus YFP quenching events was greatly reduced from ∼80% for wild-type GPc to 10% for the K33A mutant (Fig 6C).

We also tested if exposure to pH below 5.0, which has been reported to partially rescue cell-cell fusion induced by Junin GPc containing substitutions at the K33 site [49], can bypass the viral membrane permeability block imposed by the K33A mutation in LASV GPc. Exposure of LASVpp to a pH 4.0 or pH 4.5 buffer did increase the fraction of fast YFP quenching events compared to pH 5.0 treated pseudoviruses (Fig 6C). However, the ratio of fast *vs* slow YFP quenching events for the K33A mutant remained much lower than for WT GPc,

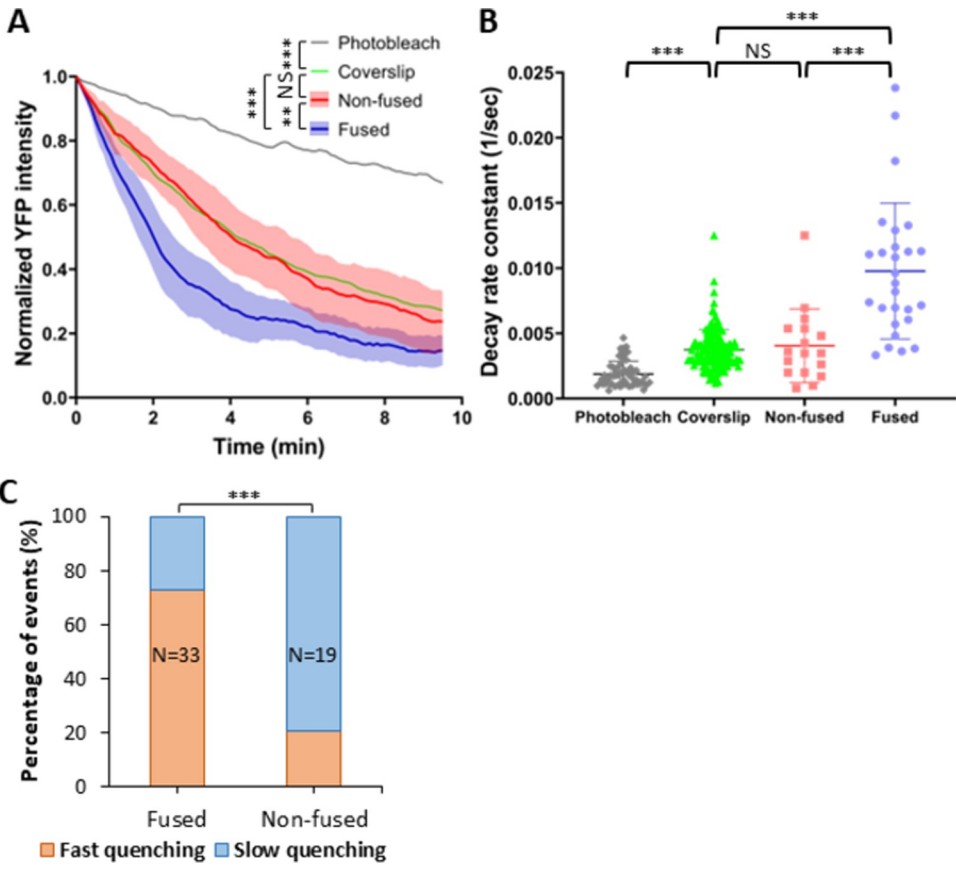

**Fig 5. Membrane permeability increases are observed for fusing but not non-fusing LASVpp.** (A) Normalized average single YFP-Vpr intensity profiles of LASVpp upon exposure to low pH on a coverslip or upon entry into A549 cells through a conventional endocytic pathway (fused vs. non-fused particles). Individual YFP intensity decays are aligned at the onset of quenching. Photobleaching-mediated YFP signal decay in PBS$^{+/+}$ is shown as reference. Shadowed area is the 95% confidence interval. Data were analyzed by two-way ANOVA. **, $p<0.01$; ***, $p<0.001$; NS, not significant. (B) The YFP-Vpr intensity decay rates for fusing vs non-fusing particles, as well as particles on a coverslip obtained by single-exponential fitting are shown. The rate of YFP photobleaching is also shown. Data were analyzed by Student's t-test. ***, $p<0.001$. (C) Quantification of different types of YFP-Vpr quenching of fused or non-fused single LASVpp in live cell analyzed by Fisher's exact test.

demonstrating that defect in membrane-permeabilizing activity of mutant GPc cannot be effectively bypassed by more acidic pH.

We also tested the effect of small molecule LASVpp fusion inhibitor, ST-193, which is thought to prevent early GPc conformational changes induced by low pH [50,51]. ST-193 abrogated LASVpp fusion with A549 cells (Fig 6D) and inhibited the decay of average YFP-Vpr signal at low pH (Fig 6E). Accordingly, single particle tracking demonstrated a greatly reduced fraction of fast YFP-Vpr quenching events for ST-193-treated viruses compared to an untreated control (Fig 6F). A correlation between LASVpp fusion activity and fast YFP-Vpr quenching implies that PVMP is associated with on-path conformational changes in GPc. Again, slow YFP quenching events likely reflect background proton permeability. Together, these results suggest that PVMP is intrinsic to GPc-mediated membrane fusion and highlight the relationship between fusion competence of GPc and PVMP.

To extend these studies, we tested a panel of additional GPc mutants that have been shown to variously impair fusion activity. The H230Y and H230E mutations in GP1 render GPc defective in LAMP1 binding, reducing LASV infectivity but not GPc-mediated syncytia

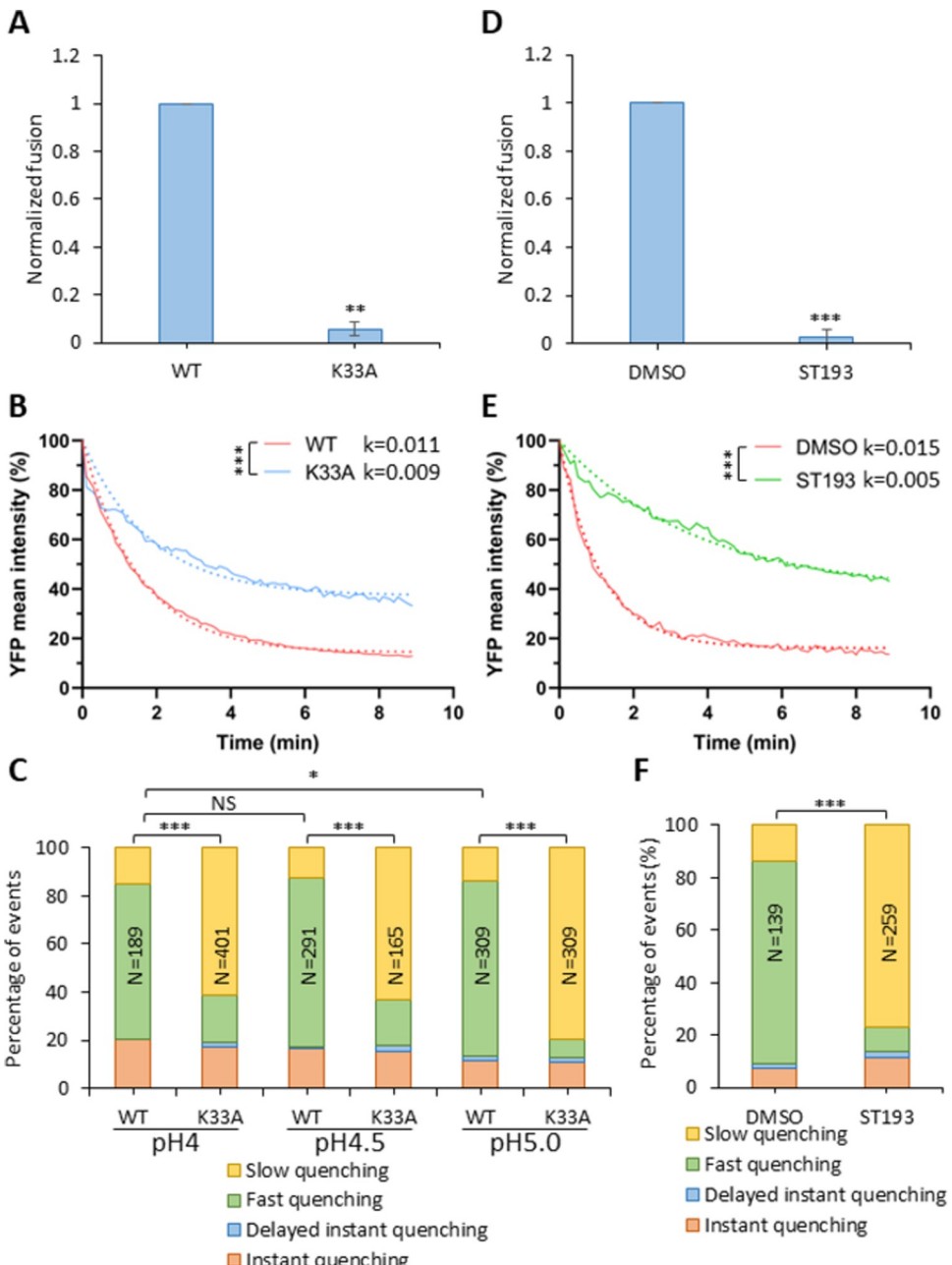

**Fig 6. Fusion-impairing LASV GPc mutation and fusion inhibitor impair viral membrane permeabilization.** (A) Wild-type and K33A mutant LASVpp-BlaM fusion with A549 cells. LASVpp was bound to A549 cells in the cold and viral fusion was initiated by shifting to 37˚C and incubating for 2 hours. Data shown are means ± SD of 3 independent experiments. Results were analyzed by Student's t-test. **, p<0.01. (B) Mean YFP-Vpr intensity decay of LASV GPc WT and K33A mutant on the surface of A549 cells after applying membrane-impermeable pH 5.0 citrate buffer. K33A mutant abrogate LASVpp fusion. The exponential decay rates k are in 1/sec. Results were analyzed by Student's t-test, ***, p<0.001. (C) Quantification of different types of single LASVpp YFP-Vpr quenching events for LASVpp WT or K33A GPc mutant on A549 cells after applying a pH 4.0, 4.5 or 5.0 citrate buffer at 37˚C. (D) ST-193 inhibits LASVpp-BlaM fusion with A549 cells. LASVpp was bound to A549 cells in the cold and viral fusion was initiated by shifting to 37˚C and incubating for 2 hours in the presence of 10 µM ST-193 or equal volume of solvent (DMSO). Data shown are means ± SD of 3 independent experiments. Results were analyzed by Student's t-test. ***, p<0.001. (E) Mean YFP-Vpr intensity decay of LASVpp on A549 cell surface after applying pH 5.0 citrate buffer in the presence or absence of 10 µM of ST-193. Exponential decay rates k are shown in 1/sec. Results were analyzed by Student's t test, ***, p<0.001. Note that the different rates of YFP quenching for WT GPc in panels B and E are due to the presence of DMSO (vehicle) in experiments with ST-193. (F) Quantification of different types of single LASVpp YFP-Vpr quenching events on A549

cells in the presence or absence of 10 µM ST-193, after applying low pH. In panels (C) and (D), the total numbers of particles analyzed by Fisher's exact test are shown above the bars. *, p<0.05; ***, p<0.001; NS, not significant.

formation [17,52]. The D268A, L280A, R282A and I323A mutations target different domains in GP2 (fusion-peptide proximal, internal fusion loop, and heptad repeat domain 1) and each attenuates GPc fusion activity [53]. These mutant GPcs are incorporating into pseudoviruses at levels comparable the wild type (S11 Fig) and all exhibit strongly reduced fusion activity, as measured by a β-lactamase-Vpr based assay (Fig 7A). The H230E mutant that abrogates

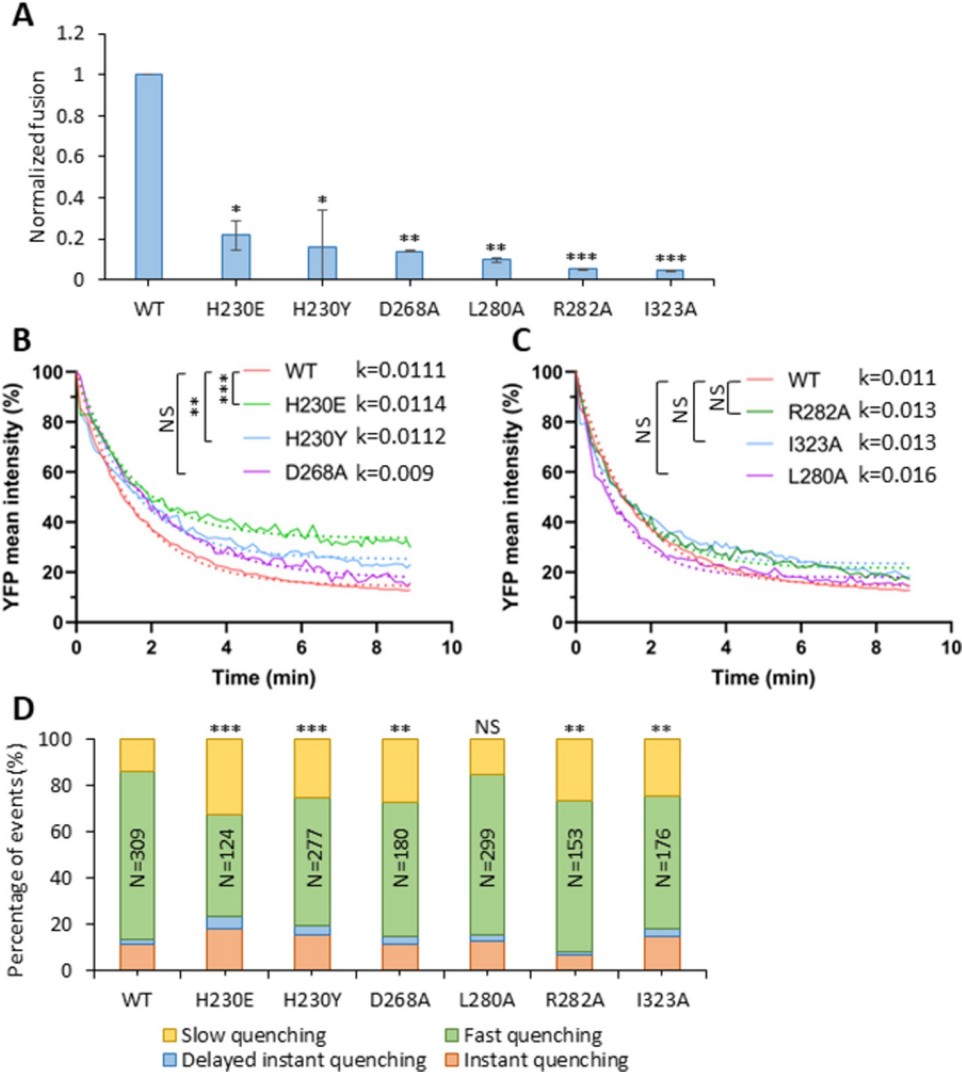

**Fig 7. LASVpp membrane permeabilization is inhibited by fusion attenuating GPc mutations.** (A) Wild-type and mutant LASVpp fusion with A549 cells measured by a BlaM assay. LASVpp was bound to A549 cells in the cold, and the fusion was initiated by shifting to 37°C and incubating for 2 hours. Data shown are means ± SD of 3 independent experiments. Results were analyzed by Student's t-test. *, p<0.05; **, p<0.01; ***, p<0.001. (B, C) Mean YFP-Vpr intensity decay of LASVpp mutants on A549 cell surface after applying membrane-impermeable pH 5.0 citrate buffer. The exponential decay rates k are shown in 1/sec. Results were analyzed by Student's t-test, **, p<0.01; ***, p<0.001; NS, not significant. (D) Quantification of different types of YFP-Vpr quenching of single LASVpp GPc mutants on A549 cells after applying low pH. Data was analyzed by Fisher's exact test. **, p<0.01; ***, p<0.001; NS, not significant.

LAMP1 binding was incorporated to a lower level compared to the wild type. Interestingly, only the two mutants defective in GP1 binding to LAMP1 significantly reduced the rate of YFP quenching upon exposure of cell-bound viruses to low pH (Fig 7B). By contrast, none of the GP2 mutations significantly affected the rate of YFP quenching (Fig 7C). In agreement with these results, the fraction of PVMP events was significantly reduced by the two H230 mutations, whereas the GP2 mutants displayed little or no effect on PVMP (Fig 7D).

Collectively, our results highlight the relationship between viral membrane permeabilization and GPc function in membrane fusion. Absent a more detailed understanding of the fusogenic conformational reorganization in GPc and the molecular impact of specific mutations and fusion inhibitors, we are unable to define a specific point within the fusion process at which PVMP is activated.

## Discussion

Viral protein-mediated membrane fusion has been extensively studied for many viruses and using a variety of experimental systems. Despite structural diversity, leading to classification of viral glycoproteins into three distinct classes [54–56], the general mechanism by which these proteins induce membrane fusion appears to be conserved. The predominant model posits that, upon triggering, viral glycoproteins undergo large-scale conformational changes involving at least two critical steps–exposure of the hydrophobic fusion peptide domain and insertion into the target membrane, followed by refolding into a hairpin-like structure that brings the fusion peptides and transmembrane domains into proximity. This latter step is thought to overcome the activation energy barrier for membrane fusion. The lipid rearrangements *en route* to membrane merger also appear to be conserved. It is widely accepted that protein-mediated membrane fusion progresses through a hemifusion intermediate, whereby the contacting outer leaflets of the virus and cell membranes are merged, but the inner leaflets remain distinct and form a hemifusion diaphragm [57–59]. Progression through a hemifusion structure is thought to prevent leakage of the cell/viral content into the external milieu. In agreement with the "tight" fusion model, our methodologies routinely find that membrane fusion by many viral glycoproteins (IAV HA, VSV G and ALSV Env) occurs without compromising the barrier function of the viral membrane, i.e., we detect no quenching of an intraviral pH-sensor in acidic endosomes prior to virus fusion [21–31] (but see [60–62] for reports of "leaky" fusion by influenza HA). In contrast, single pseudovirus fusion mediated by LASV GPc occurs after permeabilization of the viral membrane [31,32].

Our results strongly suggest that arenaviral PVMP is related to fusion-inducing conformational changes in GPc. Firstly, PVMP appears unique to arenaviruses. Increases in membrane permeability are observed with Old- and New-World arenavirus GPcs, in both HIV-1 pseudovirions and arenavirus VLPs, but not with the glycoproteins of other viruses, such as HIV-1 or IAV. PVMP is also independent of the specific target cell line. Secondly, an increase in membrane permeability at low pH is selectively observed for LASVpp that successfully fuse and rarely for particles that, for unknown reasons, do not undergo fusion. Thirdly, mutations that impair LASV fusion activity and small-molecule fusion inhibitors both interfere with the GPc's ability to induce PVMP at low pH. Fourthly, virus-cell contact greatly enhances the viral membrane permeability increase at low pH, as compared to cell-free viruses. Together, our results imply that functional conformational changes in GPc that are triggered by endosomal acidification, as well as unknown interactions of GPc with host cell proteins or membrane, are required for the PVMP.

The observation that YFP-Vpr quenching occurs shortly after the initial acidification of endosomal lumen (Fig 3E) suggests that LASV GPc conformational changes associated with

PVMP occur in early endosomes. As LASV is thought to bind LAMP-1 and complete membrane fusion in late endosomes/endolysosomes [8,18,19,63], we suggest that the permeabilized virions are subsequently trafficked to these compartments. In DF-1 cells expressing human LAMP1, the average lag time between YFP-Vpr quenching and nascent LASVpp fusion pore formation is ∼8 min [32].

Analysis of different fusion-impaired mutants suggests that mutations interfering with early conformational changes of LASV GPc (e.g., K33A [49]) tend to more potently inhibit PVMP than those affecting later steps of in the process (e.g., fusion peptide insertion into the host cell membrane or formation of the post-fusion trimer of hairpin structure). Although ST-193 was reported to also act early in pH-induced fusion activation, recent observations support an alternative interpretation that ST-193 instead arrests fusion at the hemifusion intermediate [7]. Since suboptimal conditions for triggering viral fusion tend to favor dead-end hemifusion (e.g., [64]), we propose that ST-193 inhibits PVMP, while allowing off-path conformational changes in GPc leading to a non-productive hemifusion state. Although GPc competence to successfully complete fusion appears to be required for PVMP, we are unable to identify the specific fusogenic conformational changes required for membrane permeabilization. A more detailed understanding of the molecular basis for fusion inhibition by select mutations and fusion inhibitors is needed to pinpoint the specific events leading PVMP.

Irrespective of the mechanism, PVMP is seen in both NW and OW GPc and appears to be associated with fusogenic structural changes in GPc, suggesting that acidification of the virus interior plays a role in arenavirus entry. Indeed, a recent report has shown that Z protein oligomerization is promoted by low pH [65], consistent with the notion that acidification of the virus interior can alter the virion matrix and prime the viral membrane for fusion. Furthermore, the same study finds that Z interaction with NP is also disrupted at acidic pH. We propose that arenavirus PVMP plays a role in GPc fusion and/or post-fusion steps of infection, including viral ribonucleoprotein release into the cytosol. Indeed, this notion is exemplified by the Influenza virus M2 viroporin, which is essential for uncoating of the influenza virus nucleocapsid [33,34]. This phenomenon may be even more widespread among viruses. Winter *et al.* have recently reported that passive acidification of the Ebola virus interior induces profound structural changes in the virion including the dissociation of the VP40 matrix layer from the viral membrane [66]. Further studies are needed to define the molecular mechanism for arenavirus PVMP and its role in its role in the early steps of arenavirus infection.

## Materials and methods

### Plasmids, cell lines and reagents

The plasmids pR9ΔEnv, mCherry-2xCL-YFP-Vpr, pMM310 BlaM-Vpr, pcRev, EcpH-ICAM, pCAGGS-WSN-HA, pCAGGS-WSN-NA and VSV-G have been described previously [20,22,67,68]. The LASV-GPc plasmid was a gift from Dr. François-Loic Cosset (Universite´ de Lyon, France) [69]. LASV-GPc-K33A, JUNV-GPc, JUNV GPc, NP and Z (all Candid-1 strain) were constructed by Jack Nunberg. The PolI-S Can reverse genetics S-RNA of Candid-1 JUNV plasmid was a gift from Dr. Juan Carlos de la Torre (The Scripps Research Institute) [70]. LASV-GPc-D268A, LASV-GPc-L280A, LASV-GPc-R282A and LASV-GPc-I323A were described previously [53]. MACV-GPc was a gift from Dr. Michael Farzan (Scripps Research Institute) [71]. LCMV-GPc was a gift from Shan-Lu Liu (The Ohio State University). pCAGGS-HXB2 was provided by Dr. James Binley (Torrey Pines Institute, CA) [72]. pDisplay-pHuji was from Addgene (Cat# 61556). Gag-EcpH was constructed by Dr. Kosuke Miyauchi. EcpH was amplified by PCR and replaced the mCherry in pcDNA3.1zeo+HIV Gag-mCherry by *BamHI/XhoI* restriction enzyme digestion and ligation.

To label the JUNV-NP (Candid-1 stain), YFP was inserted at an exposed loop by replacing the D93 residue with the YFP sequence [38]. The inserted YFP lacks the initiation and termination codons and is flanked by GSSG linkers. The YFP sequence was amplified by PCR (forward primer: 5' CCA TGA GGA GTG TTC AAC GAA ACA CAG TTT TCA AGG TGG GAA GCT CCG GCG TGA GCA AGG GCG AGG AGC TG; reverse primer: 5' GGT CAG ACG CCA ACT CCA TCA GTT CAT CCC TCC CCA GGC CGG AGC TGC CCT TGT ACA GCT CGT CCA TGC CGA G) and the resulting megaprimer was inserted at the D93 site (replacing D93 residue) in the PolI-S Can plasmid by QuikChange mutagenesis (Agilent Technologies, Santa Clara, CA). NP-DYFP then was adapted and extracted from the modified PolI-S Can plasmid by PCR (forward primer: 5' CGC GGC TAG CTC TGG CAT GGC ACA CTC CAA GGA GGT TCC AAG C; reverse primer: 5' CGC GCT CGA GTG CTT ACA GTG CAT AGG CTG CCT TCG G) and inserted into pcDNA3.1(+) by *NheI/XhoI* restriction enzyme digestion and ligation. The NP-DYFP insert was then transferred to the plasmid pCAGGS by *SacI/XhoI* restriction enzyme digestion and ligation.

To obtain JUNV-NP-DmCherry (Candid-1 strain), mCherry sequence was amplified by PCR (forward primer: 5' CCA TGA GGA GTG TTC AAC GAA ACA CAG TTT TCA AGG TGG GAA GCT CCG GCA TGG TGA GCA AGG GTG AGG AGG; reverse primer: 5'GGT CAG ACG CCA ACT CCA TCA GTT CAT CCC TCC CCA GGC CGG AGC TGC CCT TGT ACA GCT CGT CCA TGC CGC C) and inserted and replaced YFP in pCDNA3.1(+)-JUNV-NP-DYFP by QuickChange PCR with the mCherry segment served as megaprimers. The NP-DmCherry insert was then transferred to the plasmid pCAGGS by SacI/XhoI restriction enzyme digestion and ligation.

Human embryonic kidney HEK293T/17 cells, human lung epithelial A549 cells, human cervix epithelial HeLa cells, human bone U2OS cells, African green monkey kidney VeroE6 and fibroblast CV-1 cells, as well as chicken embryonic fibroblast DF-1 cells, were obtained from ATCC (Manassas, VA, USA). All cells were grown in high glucose Dulbecco's Modified Eagle Medium (DMEM; Mediatech, Manassas, VA, USA) containing 10% heat-inactivated Fetal Bovine Serum (FBS; Atlanta Biologicals, Flowery Branch, GA, USA) and 1% penicillin/streptomycin (GeminiBio, West Sacramento, CA, USA). The growth medium for HEK 293T/17 cells was supplemented with 0.5 mg/ml G418 (Genesee Scientific, San Diego, CA, USA).

ST-193 was purchased from MedChemExpress (NJ, USA). BSA and Bafilomycin A1 were acquired from Sigma-Aldrich (MO, USA). The recombinant human α-dystroglycan (rhDG) protein was acquired from R&D Systems (MN, USA). The LAMP1 (H4A3) mouse antibody (LAMP1ab) was purchased from Santa Cruz Biotechnology (TX, USA).

## Pseudovirus and virus like particle production

The pseudoviruses and virus like particles (VLPs) were produced by transfecting HEK293T/17 cells using JetPRIME transfection reagent (Polyplus-transfection, Illkirch-Graffenstaden, France). HEK293T/17 cells were seeded in DMEM with 10% FBS the day before transfection. To produce the pseudoviruses co-labeled with YFP and mCherry for single virus fusion experiments, HEK293T/17 cells were transfected with pR9ΔEnv, mCherry-CL-YFP-Vpr, pcRev and different envelope glycoprotein plasmids: LASV-GPc wild-tpe or mutants, MACV-GPc, JUNV-GPc, LCMV-GPc, HIV-HXB2 Env and IAV WSN HA and NA. These viral envelope encoding plasmids were omitted when producing "bald" control pseudoviruses. To produce immature LASV pseudovirus (LASV-SQV), HEK293T/17 cells were transfected with the above mixture of plasmids and maintained in the presence of 300 nM of the HIV protease inhibitor Saquinavir (SQV, contributed by DAIDS/NIAID, NIH HIV Reagent Program). To produce LASV and IAV pseudoviruses co-labeled with the surface and core pH-sensors for

single virus internalization experiments, HEK293T/17 cells were transfected with LASV-GPc or IAV WSN HA and NA, pR9ΔEnv, pDisplay-pHuji, HIV-1 Gag-EcpH, and pcRev.

To produce VLPs with YFP and mCherry labeling for single VLP fusion experiments, HEK293T/17 cells were transfected with plasmids encoding the following Candid-1 strain of JUNV-derived constructs: NP-DYFP, NP-DmCherry, Z and GPc glycoproteins of (LASV, JUNV, Candid-1, and MACV. To produce VSV-VLPs containing YFP and mCherry labeled NP and β-lactamase, HEK293T/17 cells were transfected with plasmids encoding the following Candid-1 strain constructs: NP-DYFP, NP-DmCherry, NP-BlaM, Z and VSV-G. To produce LASV pseudoviruses carrying β-lactamase-Vpr (BlaM-Vpr) for a bulk virus-cell fusion assay [46,47], HEK293T/17 cells were transfected with wild-type or mutant LASV-GPc, pR9ΔEnv, BlaM-Vpr and pcRev plasmids. Supernatants were collected at 48 hours post-transfection and filtered through a 0.45 μm syringe filter to remove cell debris and virus aggregates. LASVpp-BlaM viruses were concentrated ten times with Lenti-X concentrator (Clontech Laboratories, Mountain View, CA, USA), snap-frozen and stored at -80°C.

## Immunostaining for envelope glycoproteins

Viruses or VLPs were incubated with poly-L-lysine coated 8-well chambered glass coverslip at room temperature for 30 minutes, washed with PBS$^{+/+}$, and fixed with 4% paraformaldehyde (Electron Microscopy Sciences, PA, USA) at room temperature for 15 min. Paraformaldehyde was quenched by washing 5 times with 20 mM Tris in PBS$^{+/+}$. Samples were then blocked with 15% FBS in PBS$^{+/+}$ at room temperature for 2 hours. To detect LASV-GPc, LASV pseudo-viruses or VLP were incubated with 10 μg/ml of 12.1F human anti-LASV-GPc primary antibody (Zalgen Labs, MD, USA) [73] at 4°C overnight followed by staining with 2 μg/ml of secondary AlexaFluor647-conjugated Donkey anti-human IgG antibody (ThermoFisher Scientific, OR, USA) at room temperature for 45 min. To assess JUNV or Candid-1 GPc incorporation into VLPs, particles were incubated with 10 μg/ml of BE08 mouse anti-JUNV-GPc primary antibody (BEI resources) [74] at 4°C overnight, washed and incubated with 2 μg/ml of AlexaFluor647 Donkey anti-Mouse IgG (H+L) (ThermoFisher) at room temperature for 45 min. To detect MACV-GPc, VLP was adhered to poly-L-lysine coated 8-well chambered glass coverslip without fixation. The VLPs were incubated with pH 5.0 citrate buffer at 37°C for 30 min to trigger the GPc conformational change and then blocked with 15% FBS in PBS$^{+/+}$ at room temperature for 2 hours, at neutral pH. The sample was then incubated with 10 μg/ml of F100G5 mouse anti-JUNV-GPc fusion peptide antibody [75] (kindly provided by the Winnipeg lab, Special Pathogens Program, National Microbiology Laboratory, Public Health Agency of Canada, Winnipeg, Manitoba, Canada) at 4°C overnight, followed by incubation with 2 μg/ml of AlexaFluor647 Donkey anti-Mouse IgG (H+L) (ThermoFisher) at room temperature for 45 min. To detect VSV-G, VLPs were incubated with 10 μg/ml of mouse anti-VSV-G antibody (Santa Cruz Biotechnology, TX, USA) at 4°C overnight, followed by labeling with 2 μg/ml of AlexaFluor647 Donkey anti-Mouse IgG (H+L) (ThermoFisher) at room temperature for 45 min. To detect HIV-1 HXB2 Env, pseudoviruses were incubated with 5 μg/ml of 2G12 human anti-Env primary antibody at 4°C overnight and 2 μg/ml AlexaFluor647 donkey anti-Human IgG secondary antibody at room temperature for 45 min. To detect IAV HA, pseudoviruses were incubated with Rabbit anti-IAV WSN HA R2376 polyclonal antibody (1:100 diluted, a gift of Dr. David Steinhauer) at 4°C overnight and 2 μg/ml AlexaFluor647 Donkey anti-rabbit IgG secondary antibody (ThermoFisher) at room temperature for 45 min. Samples were washed 5 times with 15% FBS/PBS$^{+/+}$ after incubation with both primary and secondary antibodies. Images were acquired on a DeltaVision microscope using Olympus 60x UPlanFluo

/1.3 NA oil objective (Olympus, Japan). Co-localization and immunofluorescence were analyzed with the ComDet plugin in ImageJ.

## Single virus and VLP imaging in live cell

Target cells were seeded the day before imaging in 35 mm collagen coated glass-bottom Petri dishes (MatTek, MA, USA) in Fluorobrite DMEM (Life Technologies Corporation, NY, USA) containing 10% FBS, penicillin, streptomycin, and L-glutamine. Sixty μl of pseudoviruses or VLPs diluted with $PBS^{+/+}$ (30-120-fold, depending on the virus' concentration) were bound to the cells by centrifugation at 1550xg for 20 min, 4°C. Unbound viruses or VLPs were removed by washing with cold $PBS^{+/+}$. Virus or VLP entry and fusion were initiated by adding 2 ml of warm Fluorobrite DMEM supplemented with 10% FBS and 20 mM HEPES (pH 7.2). Samples were placed on a DeltaVision Elite microscope equipped with a temperature, humidity and $CO_2$-controlled chamber and immediately imaged in a time-lapse mode for 2 h. Every 6 sec, 4 Z-stacks spaced by 1.5 μm were acquired to span the thickness of cells using Olympus 60x UPlanFluo /1.3 NA oil objective (Olympus, Japan).

For particle tracking and analyses, 3D images were converted to maximum intensity projections. Single particle fusion or pH-sensitive probe quenching events were annotated with the ROI manager tool in ImageJ. Single particle tracking was performed using ICY image analysis software (icy.bioimageanalysis.org). Briefly, pseudoviruses or VLPs were identified by the Spot Detection plugin and tracked using the Spot Tracking plugin to determine their trajectories and changes in fluorescence intensity over time.

## Single virus *in vitro* acidification

To assess changes in membrane permeability resulting in acidification of virus' interior, viruses were bound to poly-L-lysine coated 8-well chambered glass coverslip at 4°C for 30 min, washed with cold $PBS^{+/+}$ to remove unbound viruses and kept in 100 μl $PBS^{+/+}$. Time-lapse images with a time interval of 2 seconds were acquired on a DeltaVision microscope using Olympus 60x UPlanFluo /1.3 NA oil objective at 37°C for 12 min. To lower the pH, 700 μl of warm membrane-impermeable pH 5.0 citrate buffer was added to each well 20 seconds after the onset of imaging.

To monitor the membrane permeability of viruses attached to the cell surface, cells were seeded in 35 mm collagen coated glass-bottom Petri dishes in Fluorobrite DMEM containing 10% FBS, penicillin, streptomycin, and L-glutamine the day before imaging. Viruses were bound to the cells by centrifugation at 1550xg for 20 min, 4°C. Unbound viruses were washed off with cold $PBS^{+/+}$ and covered with 70 μl of $PBS^{+/+}$. Time-lapse imaging was performed on a DeltaVision microscope for 12 min using Olympus 60x UPlanFluo /1.3 NA oil objective. Every 8 sec, 4 Z-stacks spaced by 1.5 μm were acquired to cover the thickness of cells. At 20 seconds after starting imaging, 500 μl of a warm pH 5.0, 4.5 or 4.0 citrate buffer was added to the wells to lower the pH. Maximum intensity projections of 3D images were used for subsequent analysis.

To analyze the YFP mean intensity change over time in low pH buffer, particles were detected, and their YFP intensity was measured by the ComDet plugin in ImageJ with mCherry as a reference color. The YFP mean intensity were calculated in Excel and fitted to a single-exponential decay model in GraphPad Prism.

To monitor membrane permeabilization and virus interior acidification at the single virus level, particles were tracked using the ICY software. Labeled pseudoviruses were identified by Spot Detection plugin and tracked using Spot Tracking plugin to determine the fluorescence intensity over time. Based on the single particle intensity change, YFP quenching was divided

into four types: instant quenching, delayed instant quenching, fast quenching and slow quenching. Instant quenching is defined as particles with YFP-Vpr signal dropping to the background level immediately (≤16 sec) upon adding a low pH buffer. Delayed instant quenching is defined as YFP-Vpr signal quickly dropping after a lag of more than 30 sec following low pH exposure.

Slow YFP quenching, likely resulting from the baseline proton permeability of the viral membrane, was defined based upon gradual quenching of YFP in coverslip-adhered pseudo-viruses that do not have viral glycoprotein responsive to low pH. The average slow YFP quenching curve was used to define slow vs fast individual quenching events based upon the intensity traces, respectively, falling within or outside the standard deviation range of the mean YFP loss curve.

## Soluble LAMP1 expression and purification

Soluble LAMP1 (sLAMP1) expression and purification was described previously [32]. Briefly, Expi293F cells were transfected with pHLsec-LAMP1 fragment, a kind gift of Juha T. Huisko-nen (University of Oxford) and incubated for three days at 37°C in the presence of 1 μg/ml of kifunensine (R&D Systems, MN, USA). Supernatant was collected and combined with half a volume of binding buffer (25 mM HEPES, 150 mM NaCl, pH 7.2). His-tagged soluble LAMP1 fragment was purified by Ni-NTA affinity chromatography. The elution was desalted and concentrated to 5 mg/ml.

## Virus-cell fusion assay

A549 cells were seeded in phenol red-free DMEM with 10% FBS the day before infection. BlaM-Vpr containing HIV-1 particles (50 pg p24) pseudotyped with wild-type or GPc were bound to cells by centrifugation at 1550xg for 30 min, 4°C. Cells were washed with cold phenol red-free DMEM with 10% FBS and 20 mM HEPES (GE Healthcare Life Sciences) to remove unbound viruses and incubated at 37°C to initiate virus-cell fusion. Cells were then placed on ice, loaded with the CCF4-AM BlaM substrate (Life Technologies), and incubated at 11°C overnight to allow substrate cleavage. The cytoplasmic BlaM activity (ratio of blue to green fluorescence) was measured using a Synergy H1 plate reader (Agilent Technologies, Santa Clara, CA, USA).

## Supporting information

**S1 Fig. IAVpp fusion with A549 cell.** (A) Illustration of mCherry-CL-YFP-Vpr labeled single IAVpp fusion. IAVpp is internalized and trafficked to acidic endosomes where it fuses with the endosomal membrane without prior membrane permeabilization (YFP quenching). IAVpp-endosome fusion results in mCherry release into the cytoplasm. (B) A single IAVpp fusion event in A549 cell. Time-lapse images (top) and fluorescence traces (bottom) show virus fusion (mCherry loss) at 1.6 min (see S2 Movie).
(TIF)

**S2 Fig. Single JUNVpp and LCMVpp fusion in HeLa cell.** (A) A representative single JUNVpp fusion event in HeLa cell. Time-lapse images (top) and fluorescence traces (bottom) show YFP quenching at 68.0 min and fusion (YFP dequenching and mCherry loss) at 77.1 min. (B) A representative single LCMVpp fusion event in HeLa cell. Time-lapse images (top) and fluorescence traces (bottom) show YFP quenching at 8.8 min and fusion (YFP dequenching and mCherry loss) at 14.9 min. A slightly delayed mCherry release after YFP dequenching

in A and C is due to a slower dilation of fusion pores to sizes that allow mCherry release.
(TIF)

**S3 Fig. Analysis of GPc incorporation into arenavirus VLPs and VLP-cell fusion.** (A) Fusion efficiency (% of cell bound-particles that fused) of single Candid-1, LASV, JUNV, MACV and VSV VLPs in A549 cells measured by imaging. Data shown are means ± SD of 3 independent experiments. Results were analyzed by Student's t-test. The number of total VLP particles analyzed is shown above the bars. Asterisks and NS on the top of bar represent the significance relative to the single Candid-1 VLP fusion efficiency in A549 cells. *, $p < 0.05$; NS, not significant. (B) Kinetics of fusion of Candid-1, LASV, JUNV and MACV GPc VLPs. (C) VSV-G VLP fusion measured by BlaM assay. (D) Images of VLPs labeled with NP-DYFP/ NP-DmCherry. VLPs were bound to poly-L-lysine coated coverslips, fixed, and incubated with antibodies against corresponding viral envelope proteins, followed by immunostaining with antibodies specific to the GPc panel. The mCherry marker of VLPs was used to identify VLP particles and visualize the associated envelope protein signal. (E) Quantification of co-localization of envelope protein immunofluorescence with viral particles identified by mCherry fluorescence. The numbers on the bars are the total numbers of VLP particles analyzed. Data shown are means ± SD of 4 imaging fields.
(TIF)

**S4 Fig. Nearly all LASVpp limit diffusion of protons through the virion membrane.** (A) Images of coverslip-adhered LASVpp in PBS (left) and 16 seconds after applying membrane-impermeable citrate pH 5.0 buffer (right). (B) Normalized YFP intensity of the only two particles that undergo instant YFP quenching and two representative particles that limit proton diffusion across their membranes. The point of adding a low pH buffer is marked with a red arrow. Tracked particles in (B) are marked by color-matched arrows in (A). Instant YPF quenching events constitute 3.9% of all particles. As shown in Fig 4, the YFP signal from the rest of single pseudovirions gradually decays over the course of several minutes (not noticeable on a short time scale shown in panel B), due to a baseline proton diffusion or due to increased membrane permeability associated with GPc refolding.
(TIF)

**S5 Fig. LASV-GPc efficiently incorporates into HIV-1 particles irrespective of virus labeling approaches.** (A) Images of LASVpp labeled with mCherry-YFP-Vpr or pHuji/Gag-EcpH. Pseudoviruses were bound to poly-L-lysine coated coverslips, fixed, and incubated with anti-LASV GPc human antibody, followed by staining with anti-human AF647-conjugated antibody. Viral mCherry or EcpH markers were used to identify viral particles and visualize the associated GPc signal. (B) Quantification of co-localization of LASV GPc immunofluorescence with viral particles identified by mCherry or EcpH fluorescence, as indicated. The numbers of LASVpp analyzed are shown above the bars. Data shown are means ± SD of 4 imaging fields. (C) Cumulative distributions of the GPc immunofluorescence intensities for each virus preparation.
(TIF)

**S6 Fig. Time lapse images at indicated time points after adding low pH buffer to viruses on coverslips and on cells.** YFP fluorescence quenching at low pH is readily detectable, while the reference mCherry signal is largely unchanged. The apparent loss of mCherry puncta on A549 cells (lower panel) is due to cell shrinkage at low pH which results in a fraction of particles moving out of focus.
(TIF)

**S7 Fig. Examples of four types of YFP-Vpr quenching of single LASVpp on coverslip (A-D) and on DF-1 cells (E-H).** The time points when low pH buffer was added are marked with red arrows. See Fig 4.
(TIF)

**S8 Fig. Estimation of viral membrane permeability to protons.** Representative pseudo-viruses exhibiting fast and slow YFP quenching were used to estimate their membrane permeability to protons using the equation $P = k*V/S$, where k is the permeability coefficient determined by exponential fit of normalized YFP fluorescence decay (assuming that fluorescence changes reflect changes in intraviral proton concentration), V is the volume and S is the surface area of the virus. The P values shown are for slow and fast quenching events assuming the particle radius is 60 nm and that the enclosed inner volume is "empty", which likely results in overestimation of the permeability value. Nonetheless, the obtained P values are much lower than those reported in the literature for liposomes. It is thus possible that the viral interior proteins buffer the inner pH by binding the incoming protons and slowing down YFP quenching.
(TIF)

**S9 Fig. Viral glycoproteins efficiently incorporate into HIV-1-based pseudoviruses.** (A) Images of the pseudoviruses labeled with mCherry-CL-YFP-Vpr. Pseudoviruses were bound to poly-L-lysine coated coverslips, fixed, and incubated with anti- LASV-GPc, anti-HIV Env or anti-IAV HA antibodies, as indicated in the figure. Viruses then were stained with respective AF647-conjugated secondary antibodies. Viral particles were identified based on the mCherry signal. Bald particles lacking envelope glycoproteins and GPc-containing viruses produced in the presence of saquinavir viruses were used as controls. (B) Quantification of co-localization of viral glycoproteins detected by immunofluorescence signal and viral particles labeled with mCherry. The numbers of LASVpp analyzed are shown above the bars. Data shown are means ± SD of 4 imaging fields. (C) Distribution of GPc immunofluorescence intensity for different pseudoviruses.
(TIF)

**S10 Fig. LASV receptors, α-dystroglycan and LAMP1, do not promote LASVpp membrane permeabilization.** LASVpp were bound to the cell surface or to poly-L-lysine coated coverslips at 4˚C. GPc conformational changes are triggered by applying membrane impermeable pH 5.0 citrate buffer. (A) For soluble LAMP1 (sLAMP1) treatment, 200 μg/ml of sLAMP1 was added to viruses in a pH 5.0 citrate buffer. For LAMP1 antibody (LAMP1ab) treatment, cells were incubated with the growth medium containing 100x diluted LAMP1ab for 1 hour before imaging; LAMP1ab was also present in the pH 5.0 citrate buffer. Two hundred μg/ml of BSA was in citrate pH 5.0 buffer used as a control. (B) For recombinant human α-dystroglycan (rhDG) treatment, LASVpp were incubated with 50 μg/ml of rhDG or BSA (control) at 37˚C for 20 min, the pH 5.0 citrate buffer was supplemented with 50 μg/ml of BSA or rhDG. The rate constants k are in 1/sec.
(TIF)

**S11 Fig. Mutant LASV-GPc efficiently incorporate into LASVpp.** (A) Images of the pseudo-viruses labeled with mCherry-CL-YFP-Vpr. Pseudoviruses were bound to poly-L-lysine coated coverslips, fixed, and incubated with anti-LASV GPc human antibody, followed by staining with AF647-conjugated anti-human antibody. Viruses were identified based on the mCherry marker. (B) Quantification of the co-localization of GPc immunofluorescence and viral particles identified by mCherry. The numbers of LASVpp analyzed are shown above the bars. Data shown are means ± SD of 4 imaging fields. (C) Distribution of the GPc immunofluorescence

intensities for each virus.
(TIF)

**S1 Movie. Single LASVpp fusion with A549 cells manifested in sequential YFP quenching and dequenching.** Single LASVpp exhibits YFP quenching at 12.3 min followed by YFP dequenching/mCherry loss at 25.6 min corresponding to virus interior acidification and fusion, respectively. Movie is related to Fig 1B.
(MP4)

**S2 Movie. Single IAVpp fusion with A549 cells manifested in mCherry release.** Single IAVpp exhibits mCherry loss at 1.6 min corresponding to virus fusion. Movie is related to S1B Fig.
(MP4)

**S3 Movie. Single Candid-1 VLP fusion with A549 cells manifested in sequential YFP quenching and dequenching.** Single Candid-1-VLP exhibits YFP quenching at 12.1 min followed by YFP dequenching at 33.4 min corresponding to VLP interior acidification and fusion, respectively. White arrows marked the VLP of concern. Movie is related to Fig 2B.
(MP4)

**S4 Movie. Single LASVpp enters acidic compartments and undergoes membrane permeabilization in A549 cells.** Single LASVpp exhibits pHuji quenching at 107.6 min followed by EcpH quenching at 108.2 min corresponding to virus entering acidic endosome and interior acidification, respectively. Movie is related to Fig 3B.
(MP4)

**S5 Movie. Single IAVpp enters acidic compartments without membrane permeabilization in A549 cells.** Single IAVpp exhibits pHuji quenching at 23.9 min corresponding to virus entering acidic endosome. Movie is related to Fig 3C.
(MP4)

## Acknowledgments

The authors wish to thank Drs. Cohen-Dwashi and Ron Diskin for certain LASV GPc mutants, Dr. Kosuke Miyauchi for cloning the Gag-EcpH construct, Zalgen Labs for a kind gift of 12.1F anti-Lassa GPc antibody. We are indebted to the Melikian lab members, Mariana Marin, Gokul Raghunath, Manish Sharma, Monica Cortez, and David Prikryl for critical reading of the manuscript and helpful suggestions.

## Author Contributions

**Conceptualization:** You Zhang, Melinda A. Brindley, Jack H. Nunberg, Gregory B. Melikyan.

**Formal analysis:** You Zhang, Gregory B. Melikyan.

**Funding acquisition:** Gregory B. Melikyan.

**Investigation:** You Zhang, Joanne York, Melinda A. Brindley.

**Methodology:** Jack H. Nunberg.

**Resources:** Melinda A. Brindley, Gregory B. Melikyan.

**Supervision:** Jack H. Nunberg, Gregory B. Melikyan.

**Writing – original draft:** You Zhang, Gregory B. Melikyan.

**Writing – review & editing:** Joanne York, Melinda A. Brindley, Jack H. Nunberg,
Gregory B. Melikyan.

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
