## [Decision Letter · Decision Letter 0]

10 Apr 2023

Dear Dr. Melikyan,

Thank you very much for submitting your manuscript "Fusogenic Structural Changes in Arenavirus Glycoproteins are Associated with Viroporin Activity" for consideration at PLOS Pathogens. As with all papers reviewed by the journal, your manuscript was reviewed by members of the editorial board and by several independent reviewers. In light of the reviews (below this email), we would like to invite the resubmission of a significantly-revised version that takes into account the reviewers' comments.

Please pay special attention to the following major points raised by the reviewers: Further quantification from multiple single tracking instances is needed to validate some claims. The study would benefit from the addition of expression levels for the glycoproteins and their mutated versions (Reviewer 1); Can further efforts be made to specify when during fusion the permeabilization occurs? Is direct interaction between GPc and the endosomal membrane (or plasma membrane) necessary? (Reviewer 2); additional discussion of data shown in Fig. 5B, ST193, and sequencing curves (Reviewer 3).

We cannot make any decision about publication until we have seen the revised manuscript and your response to the reviewers' comments. Your revised manuscript is also likely to be sent to reviewers for further evaluation.

Sincerely,

Ronald N. Harty

Guest Editor

PLOS Pathogens

Matthias Schnell

Section Editor

PLOS Pathogens

Kasturi Haldar

Editor-in-Chief

PLOS Pathogens

orcid.org/0000-0001-5065-158X

Michael Malim

Editor-in-Chief

PLOS Pathogens

orcid.org/0000-0002-7699-2064

Please pay special attention to the following major points raised by the reviewers: Further quantification from multiple single tracking instances is needed to validate some claims. The study would benefit from the addition of expression levels for the glycoproteins and their mutated versions (Reviewer 1); Can further efforts be made to specify when during fusion the permeabilization occurs? Is direct interaction between GPc and the endosomal membrane (or plasma membrane) necessary? (Reviewer 2); additional discussion of data shown in Fig. 5B, ST193, and sequencing curves (Reviewer 3).

We cannot make any decision about publication until we have seen the revised manuscript and your response to the reviewers' comments. Your revised manuscript is likely to be sent to reviewers for further evaluation.

Reviewer's Responses to Questions

**Part I - Summary**

Reviewer #1: Zhang et al, in the manuscript “Fusogenic structural changes in Arenavirus glycoproteins are associated with viporin activity” use single virus and vlp tracking to investigate the effect of the Lassa virus glycoprotein (GPc) conformational changes on virion membrane permeability. Their findings suggest that virion interior acidification results in viral membrane permeation and is mediated by GPc conformational changes. GPc binding to membrane-bound LAMP1, a host receptor, is required for driving GPc conformation changes. ST-193, a GPC inhibitor previously hypothesized to block at an early conformational change, blocked GPcs at a late conformational change increasing the virion membrane permeabilization and acidification of the virus interior. The authors conclude that GPc-mediated acidification plays important roles in viral fusion including ribonucleoprotein release into the cytosol. Insight into the arenavirus GPc fusion process is important and the findings regarding GPc’s effect on membrane permeability would be beneficial to the virology community. The paper is well written and scholarly. This is an interesting and innovative story. Much of the data are strong and convincing. Some of the claims are based on the authors’ interpretation of a single event. Further quantification from multiple single tracking instances is needed to validate some of these claims. This single particle tracking method was previously published (Sood et al. 2017) with robust quantification of viral fusion events. The present study would also benefit from the addition of expression levels for the glycoproteins and their mutated versions.

Reviewer #2: Zhang and colleagues build upon the already substantial body of work by the Melikyan laboratory on Lassa virus fusion. In two previous publications, the Melikyian laboratory has reported that permeabilization of the viral membrane occurs prior to Lassa GPc-mediated membrane fusion. This phenomenon has thus been established, although the stage during viral fusion and the importance of acidification of the virion interior for post-entry events remains unknown. The present work seeks to strengthen the correlation between membrane permeabilization and fusogenic structural changes in the arenavirus GPc. The results nicely demonstrate that membrane permeabilization is a common feature of arenavirus GPcs, and not limited to Lassa virus, and that it requires interaction with the target cell membrane; binding to soluble forms of the alpha-DG or LAMP1 receptors is insufficient to induce permeabilization. The experiments are expertly performed and analyzed. Unfortunately, a clear link between membrane permeabilization and fusion (or other downstream events) remains unclear.

Reviewer #3: Review Grisha

The authors present a series of compelling experiments to elucidate a mystery, the apparent permeabilzation of the Lassa virus envelope during membrane fusion. The experiments are stunning in their clarity and the interpretations are sound. This is significant news and sound science.

The last sentence of the abstract is the most interesting, but there is little discussion or data to support this sentence. It is more conjectural as presented. The uncoating of the IAV core comes to mind as presented but I have no idea if the insides of the LASV also unwraps or hydrates or reacts at all to low pH. The analogy to M2 is worth bringing up in the introduction, since the authors have already shown that LASV has an envelope proton permeability during fusion. Then having investigated it more systematically they can make a more substantial proposal based on this systematic work.

Similarly, it is not until the very last paragraph of the paper that all the interesting points are raised, and now it is too dense. I then realize that I cannot tell if a membrane-disrupting fusion peptide can just fall back onto the envelope and cause these changes, but only when on a cell surface. Nor can I tell in WT if there is ever fusion without permeabilization, vs. permeabilization without fusion, because the aggregate data of Fig. 5B is not linked. ST193 is very important, I would suggest the authors devote a paragraph on ST193 alone.

I would have appreciated a better description of what might be expected in the curves of dequenching. There are multiple nonlinear factors that might accelerate the dequenching, yet it is constantly slowing. Perhaps something can be learned by averaging together all the dequenching curves with one condition?

On page 19, the authors say: “Our results strongly suggest that arenaviral membrane permeabilization is mediated by fusion inducing conformational changes in GPc”. I am not convinced. The experiment shows the difference between on glass and on cell. It may be that cell proteins are involved as well. Perhaps the authors think that “mediated” means “play some role” but I do not read it that way. “it is likely?” perhaps?

Stylistic issues:

I do not find the term “leaky fusion” to be very useful anymore. In the old days of viral fusion research, it was very important to distinguish “real” fusion from artefact, and “leaky” fusion was equated with artefact. But the current interest is more on pores in virology, and this is another example of leaks on one side or the other of the hemifusion diaphragm. It iseems to be important. To call it “leaky” may do the authors a disservice.

In addition, I find the paper somewhat wordy and stilted, and it obscures the beautiful experiments. I offer one example of my suggestions: Page 13 of the pdf (sorry, the reviewer’s copy did not have page numbers).

a) Cut “…exhibited distinct YFP quenching profiles and…”.

b) Shown instead of “manifested”

c) “…particles with presumably preexisiting…”

d) “…increase in the baseline viral…” could be “… increased viral…” do you mean increased envelope permeability? What does the baseline have to do with it? To my mind, the baseline is what comes before shifting to pH 4.

Page 14: middle paragraph could use a conclusion, what is the point of all these numbers?

Page 15: nine lines from bottom: I do not know enough to understand the link between the the last two sentences of this paragraph. Anyway, it seems to be a discussion point, not a result.

Page 20: last sentence of middle paragraph: I had to put brackets around “…YFP-Vpr quenching” and “pore formation” to get to the 8 min. Perhaps cut “DF-1 cells expressing human LAMP1” or get the 8 min in front?

Finally there is no structural data in this paper, just mutants implying structural changes.

Figs 1-3 and S1: The cartoon shows “Neutral endosome”. Does such a thing exist? The cartoon makes a big deal out of something but later it seems that it is delayed entry. I would fix this. It is likely that the endosome starts to acidify somewhat early after pinching off. Perhaps the authors could refer to their earlier papers if such a thing exists, and then they would have to add experiments to find the delay between pinching off, acidification, and permeabilization. Perhaps it is simpler to delete this potentially misleading cartoon part?

Fig S2: NS or ** compared to which bar? All compared to A549?

Fig S9: Title to legend: “Viral glycoproteins are…” Don’t you want to erase the word “are”?

Movies: I do not see the point of including movies, there are few pixels and the figure data shows the same information. No mention of the movements inside the endosomes are mentioned in the text. I would delete them and place into a database for data sharing. Anyway Movie S3 does not show in mac Quicktime player or in mac Fiji.

**Part II – Major Issues: Key Experiments Required for Acceptance**

Reviewer #1: 1) Figures 1, 2, and 3 rely on single viral fusion instances for their claims. Quantification of multiple instances and showing more expansive views of these interactions, as in movie S3, is needed to validate the author’s claims.

2) For the HIV-1 pseudotyped YFP-Vpr studies (Fig. 1), the authors show IAVpp does not contain the leaky virus-endosome phenotype (Suppl. Fig. S1) yet they do not seem to show this necessary control for the arenavirus VLP NP-YFP studies.

3) The authors state, “Quantitative differences in the efficiency and kinetics of VLP-cell fusion mediated by different GPcs (Suppl. Fig. S3) are likely caused by varied levels of expression of these glycoproteins or incorporation into Candid-1-based VLPs.”. A western blot or other quantitation of the relative expression levels would help to explain these differences. Similarly, the author’s claim, “… the relatively minor fraction of instant YFP quenching events (Fig. 4) does not considerably vary across the mutants…” yet these differences could be attributed to different expression levels in the absence of precise data on expression.

Reviewer #2: The major concern is regarding the extent to which the current results advance our understanding of the role of viral membrane permeabilization, or the stage of entry during which it occurs. Can further efforts be made to specify when during fusion the permeabilization occurs? Is direct interaction between GPc and the endosomal membrane (or plasma membrane) necessary? Based on the discussion in the manuscript, the authors appear to feel that permeabilization occurs prior to GPc interaction with the endosomal membrane. Perhaps evaluation of permeabilization during inhibition of Lassa virus fusion with a heptad repeat-derived peptide would help strengthen this point.

In general, it’s difficult to know what to conclude from the data collected on GPc mutants. As the authors note, the K33A mutant is thought to interrupt pH sensing. Does the K33A mutant still have the same dependence on pH for YFP quenching?

Furthermore, all the GP1 and GP2 mutants considered reduce fusion by ∼90% (or more), and yet have very modest effects, or in some cases no effect, on the kinetics of YFP quenching and the fraction of fast quenching events. How do we interpret the magnitude of these differences in effects of the mutants? The authors argue that permeabilization occurs early and the mutations inhibit fusion at a later step. But ST-193, which inhibits at the point of hemifusion (ie, a late event), has a much larger effect of the kinetics of YFP quenching. How do we reconcile these observations?

Reviewer #3: (No Response)

**Part III – Minor Issues: Editorial and Data Presentation Modifications**

Reviewer #1: 1) For Figure 7, GPc mutated protein expression levels would be helpful.

2) It would be helpful to merge figures 1 and 2 together so that comparisons between pseudotyped and virus-like particles could be made in the same figure.

3) In multiple places percentages are given, without statistical data provided to validate those percentages. For example, it is unclear what is the total number of tracked particles when the author states “… only 14% of cell-bound particles exhibited a loss of pHuji signal within 2 hrs of infection (Fig. 3D) …”.

Reviewer #2: The text refers to a L270A mutation in GP2. But Figure 7 refers to L280A.

Figure 7B indicates statistical significance in the rates of YFP quenching for WT, and H230E/Y, and yet the indicated rates are all 0.011.

The authors may be interested to note that acidic pH-induced remodeling on the Ebola VP40 matrix due to permeabilization of the viral membrane has also been recently reported (https://doi.org/10.1101/2022.08.24.505067).

Reviewer #3: (No Response)

PLOS authors have the option to publish the peer review history of their article (what does this mean?). If published, this will include your full peer review and any attached files.

Reviewer #1: No

Reviewer #2: No

Reviewer #3: No

Figure Files:

Data Requirements:

Please note that, as a condition of publication, PLOS' data policy requires that you make available all data used to draw the conclusions outlined in your manuscript. Data must be deposited in an appropriate repository, included within the body of the manuscript, or uploaded as supporting information. This includes all numerical values that were used to generate graphs, histograms etc.. For an example see here on PLOS Biology: http://www.plosbiology.org/article/info:doi%2F10.1371%2Fjournal.pbio.1001908#s5.
---

## [Decision Letter · Decision Letter 1]

4 Jul 2023

Dear Dr. Melikyan,

We are pleased to inform you that your manuscript 'Fusogenic Structural Changes in Arenavirus Glycoproteins are Associated with Viroporin Activity' has been provisionally accepted for publication in PLOS Pathogens.

Best regards,

Ronald N. Harty

Guest Editor

PLOS Pathogens

Matthias Schnell

Section Editor

PLOS Pathogens

Kasturi Haldar

Editor-in-Chief

PLOS Pathogens

orcid.org/0000-0001-5065-158X

Michael Malim

Editor-in-Chief

PLOS Pathogens

orcid.org/0000-0002-7699-2064

Reviewer Comments (if any, and for reference):

Reviewer's Responses to Questions

**Part I - Summary**

Reviewer #1: The authors have adequately responded to the criticisms but still need to address the following concern raised in the initial review:

The authors state that the full data analysis from Fig. 2 is in Suppl. Fig. S3 but this correlation cannot be found in the text or figure legends. The authors should reference the supplemental figure in the main text when referring to the figure panels. Addressing this concern will greatly aid readers unfamiliar with this work. In fact the numbers above the bars in Supp 3A should be included in the main text.

Reviewer #2: I appreciate the authors' thoughtful responses. All of my concerns have been addressed.

Reviewer #3: I am very pleased in general with the revised manuscript. The authors have responded to all of the reviewers’ comments.

1. I was not explicit enough in my comment of Reviewer #3. I stated:: “Finally there is no structural data in this paper, just mutants implying structural changes.” I should have written: I find the title of the paper misleading. It says that structural changes are the main conclusion of this paper, implying that structures will be demonstrated by this paper, implying that it is a structural biology paper. But in fact, it is not a structural biology paper, as the authors readily agree (see below). I think the title need to be altered.

The authors agree, but did not change the title:

“Response: We agree. At this stage, getting structural information on GPc intermediates and/or viral membrane defects is not technically feasible.”

2. I am a bit worried about the abbreviation “PVMP”. It signifies “pre-fusion viral membrane permeabilization (PVMP).“ Unfortunately, PVMP can also stand for “post-fusion…”

Is it possible to use the word “pre-VMP” or something else? 

3. I cannot understand the response sentence that starts “Our conclusion” Is it based or not based? It says it is based not solely based. And then it is a run-on sentence with two parenthetical phrases. What does it mean? Does it mean that it is based on two conclusions, a) and b). Perhaps it should say that?

“On page 19, the authors say: “Our results strongly suggest that arenaviral membrane permeabilization is mediated by fusion inducing conformational changes in GPc”. I am not convinced. The experiment shows the difference between on glass and on cell. It may be that cell proteins are involved as well. Perhaps the authors think that “mediated” means “play some role” but I do not read it that way. “it is likely?” perhaps?

Response: We thank the reviewer for this comment. Our conclusion is based not solely based upon the difference between coverslip-attached and cell-bound viruses, but also on the marked difference between fusing and non-fusing pseudoviruses in cells (Fig. 5), as well as the effects of fusion-impairing GPc mutations on the membrane permeability increases.

**Part II – Major Issues: Key Experiments Required for Acceptance**

Reviewer #1: (No Response)

Reviewer #2: (No Response)

Reviewer #3: (No Response)

**Part III – Minor Issues: Editorial and Data Presentation Modifications**

Reviewer #1: (No Response)

Reviewer #2: (No Response)

Reviewer #3: (No Response)

PLOS authors have the option to publish the peer review history of their article (what does this mean?). If published, this will include your full peer review and any attached files.

Reviewer #1: No

Reviewer #2: No

Reviewer #3: No

---

## [Editor Report · Acceptance letter]

21 Jul 2023

Dear Dr. Melikyan,

We are delighted to inform you that your manuscript, "Fusogenic Structural Changes in Arenavirus Glycoproteins are Associated with Viroporin Activity," has been formally accepted for publication in PLOS Pathogens.

Best regards,

Kasturi Haldar

Editor-in-Chief

PLOS Pathogens

orcid.org/0000-0001-5065-158X

Michael Malim

Editor-in-Chief

PLOS Pathogens

orcid.org/0000-0002-7699-2064